# In situ observation of oscillatory redox dynamics of copper

Jing Cao[1], Ali Rinaldi[2], Milivoj Plodinec [1], Xing Huang [1,3], Elena Willinger[1,3], Adnan Hammud[1], Stefan Hieke[4], Sebastian Beeg[5], Luca Gregoratti[6], Claudiu Colbea[3], Robert Schlögl[1,5], Markus Antonietti[7], Mark Greiner [5✉] & Marc Willinger [1,3,7✉]

How a catalyst behaves microscopically under reaction conditions, and what kinds of active sites transiently exist on its surface, is still very much a mystery to the scientific community. Here we present an in situ study on the red-ox behaviour of copper in the model reaction of hydrogen oxidation. Direct imaging combined with on-line mass spectroscopy shows that activity emerges near a phase boundary, where complex spatio-temporal dynamics are induced by the competing action of simultaneously present oxidizing and reducing agents. Using a combination of in situ imaging with in situ X-ray absorption spectroscopy and scanning photoemission microscopy, we reveal the relation between chemical and morphological dynamics and demonstrate that a static picture of active sites is insufficient to describe catalytic function of redox-active metal catalysts. The observed oscillatory redox dynamics provide a unique insight on phase-cooperation and a convenient and general mechanism for constant re-generation of transient active sites.

[1] Department of Inorganic Chemistry, Fritz-Haber-Institut der Max-Planck-Gesellschaft, 14195 Berlin, Germany. [2] Chemistry Department, King Fahd University of Petroleum & Minerals, 31261 Dhahran, Saudi Arabia. [3] Scientific Center for Optical and Electron Microscopy, ScopeMETH Zürich, 8093 Zürich, Switzerland. [4] Max-Planck-Institut für Eisenforschung GmbH, 40237 Düsseldorf, Germany. [5] Max Planck Institute for Chemical Energy Conversion, 45470 Mülheim an der Ruhr, Germany. [6] Elettra-Sincrotrone Trieste S.C.p.A, 34149 Basovizza, Trieste, Italy. [7] Max-Planck-Institute of Colloids and Interfaces, Department of Colloid Chemistry, 14424 Potsdam, Germany. ✉email: mark.greiner@cec.mpg.de; willmarc@ethz.ch

Heterogeneous catalysts interact with molecules of the surrounding gas-phase and facilitate the breaking and making of chemical bonds. During the interaction, reactant molecules can chemically alter the catalyst, and in doing so, can give rise to a dynamically changing catalyst surface, the coexistence of multiple surface phases, and even the formation of meta-stable phases[1–5]. Such situations occur when an oxidizable or reducible material is used to catalyse a redox reaction. Examples are methanol oxidation on copper[6–8], ethylene epoxidation on silver[8], or CO oxidation over palladium[9].

Surface sensitive spectroscopic methods have been applied to study the state of active metal catalysts under redox conditions and to reveal the relation between catalytic performance and the presence of different oxygen species[6,10–12]. However, despite of the detailed integral spectroscopic characterization, spatially resolved insight about the associated morphological and chemical dynamics is still scarce[13–15]. The present approach attempts to gain conceptual knowledge about the active state of a catalyst using a combination of in situ scanning electron microscopy (SEM) with chemical-state-sensitive near-ambient pressure X-ray photoemission spectroscopy (NAP-XPS) and environmental scanning photoemission microscopy (ESPEM). Hydrogen oxidation has been chosen as model redox reaction, and copper as model catalyst because of its relevance in a number of industrial redox reactions, such as methanol oxidation $(CH_3OH + O_2)$[16], water gas shift reaction[17], $CO_2$ reduction[18] and as an active component in methanol synthesis catalysts[19].

Through the combination of in situ imaging with in situ spectroscopy, we are able to correlate the gas-phase induced morphological dynamics with laterally resolved chemical information about the involved phases. It is important to note here that the recorded in situ movies are central to this paper. The reader is thus strongly advised to download and watch the movies.

## Results

**Phase diagram**. Freshly annealed and reduced samples were exposed to mixtures of $H_2$ and $O_2$ at various ratios inside the chamber of an environmental scanning electron microscope (ESEM) and heated to temperatures between 600 and 800 °C by direct illumination with an infrared laser (for details, see 'Experimental' section and Supplementary Fig. 1). During in situ SEM observation at pressures between 20 and 50 Pa, the morphological evolution of freshly annealed samples was studied as a function of temperature and gas composition (see Fig. 1a–c). Depending on the $H_2/O_2$ ratio, three distinct regimes could be identified: With the addition of small concentrations of $O_2$ to the $H_2$ flow, the initially flat surface started to reconstruct (Fig. 1a and regime A in Fig. 1d). The type of surface reconstruction clearly showed a dependence on the grain orientation and $O_2$ partial pressure. With increasing oxygen concentration, some grains developed an apparently smooth surface after passing through different surface reconstructions (see Supplementary Fig. 2). Within regime A, however, the reducing action of hydrogen dominated and no signs of oxide formation could be detected. Once the oxygen concentration was increased to about 3–5%, formation and growth of corrugated oxide islands could be observed (Fig. 1b). They can easily be distinguished from the metallic copper by their morphology and identified by energy dispersive X-ray analysis (see EDX map, Supplementary Fig. 3). However, due to the presence of hydrogen, freshly formed oxide islands were instable and faced reduction. The competing action of hydrogen and oxygen led to the establishment of a dynamic equilibrium (regime B in Fig. 1d). The latter is characterized by phase co-existence and constant inter-conversion between metal and oxide islands, such as shown in Supplementary Movies 2 and 3.

At oxygen concentrations above 20–30%, the oxidizing force is dominant (region C in Fig. 1d). As a consequence, the surface got fully covered with copper oxide. As shown in Fig. 1d, the $O_2$ concentrations at which boundaries between different regimes were observed, depend on temperature. At higher temperatures, higher $O_2$ partial pressures were required to move out of the bi-stability regime into the oxidized regime. Similarly, the $O_2$ partial pressure needed to transition from the metallic regime to bi-stability increased with temperature. The need for higher $O_2$ partial pressures with higher temperatures is expected, given that the chemical potential of oxygen decreases with increasing temperature[20–23]. On the low temperature side, the affinity of copper towards oxidation dominates. At 300 °C, we observed surface oxidation at oxygen concentrations below 1% in hydrogen. First signs of a redox regime could be detected at around 400 °C (see Supplementary Fig. 4). Simultaneously with the onset of the red-ox dynamics, the formation of $H_2O$ could be detected by quadrupole mass spectrometric analysis of the gas-composition near the sample (see Supplementary Fig. 5). Although the in situ SEM set-up was not optimized for efficient detection of the water signal, it provided sufficient information to correlate the onset of catalytic activity with the emergence of structural redox dynamics at around 400 °C. In regime B, the structural dynamics, as well as the detected amount of $H_2O$, increased with temperature. It should also be mentioned here that we did not detect any influence of the electron beam on the observed dynamics (for more information, see 'Methods' section).

**Spatio-temporal dynamics**. Redox dynamics observed across the whole regime B were characterized by the simultaneous presence of metal and oxide islands and their constant interconversion. However, the size of the growing oxide domains and the overall time-scale of structural dynamics varied with temperature and $H_2/O_2$ ratio (see Supplementary Movies 2 and 3). During the mapping of the dynamic phase diagram shown in Fig. 1d, we observed a regime in which the system spontaneously developed laterally synchronized kinetic oscillations. The existence of oscillating regimes has been reported for copper in methanol and propane oxidation[24,25]. In the oscillating state, the spatiotemporal dynamics can be disentangled and the sequential transition of the surface through different states studied. It is this regime, located at around 700 °C and a $H_2/O_2$ ratio of 96/4, on which we will focus in this work. Admittedly, this temperature is relatively high compared to typical Cu catalysed reactions. Important for catalysis in general is that the redox dynamics are present across the whole parameter regime B shown in Fig. 1 and further down to approximately 400 °C. They are thus relevant for industrial reactions that take place at lower temperature, but at substantially higher pressure. A time series of images recorded at 700 °C in an atmosphere containing 4% oxygen in hydrogen is shown in Fig. 2. The full dynamics are presented in Supplementary Movie 4 (overview) and Supplementary Movie 5 (higher magnification).

One can see that the interaction of the surface with the two counter-acting components of the gas phase induces a sequence of stages that are characterized by distinct morphologies. Starting with a heavily facetted surface of the central grain in Fig. 2a, the surface transforms into a microscopically smooth state (Fig. 2b). Thereafter, the formation of copper oxide islands and propagation of oxidation fronts across the surface is observed (Fig. 2c). The oxide islands are instable in the hydrogen-rich atmosphere and get reduced shortly after having formed. With the disappearance of the oxide islands, the facetted surface state is re-established (Fig. 2d) and one redox-cycle completed. Individual frames that were recorded from the central grain at higher magnification during a subsequent redox cycle are shown in

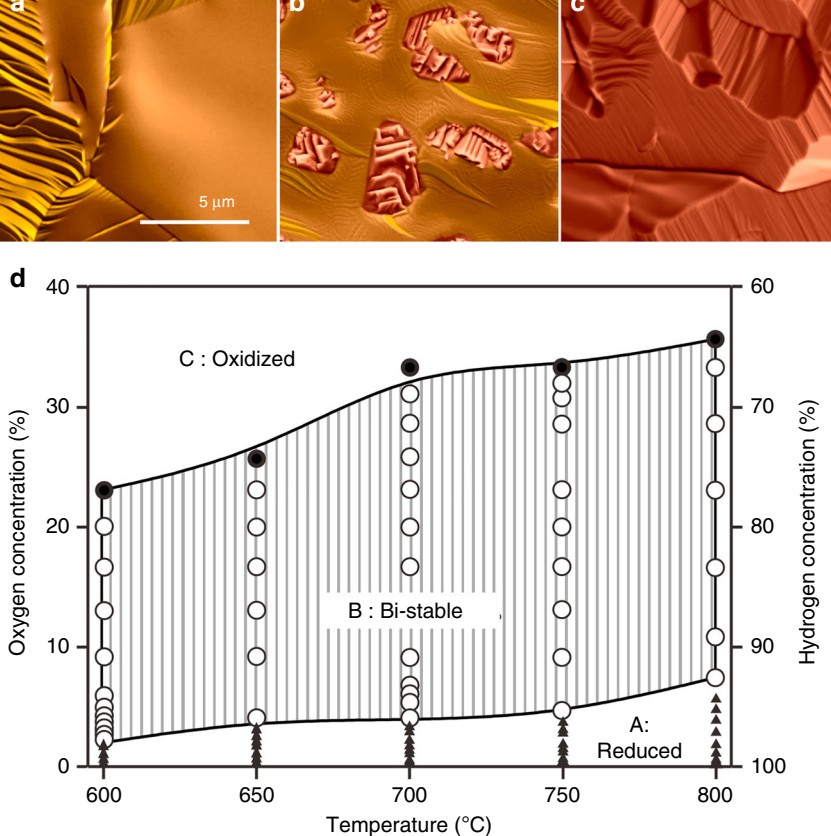

**Fig. 1 Dynamic phase diagram. a–c** SEM images showing different surface states corresponding to, respectively, regimes A, B and C of the phase diagram presented in **d**. In regime A, the reductant dominates and the catalyst remains in the metallic state. Some grains show oxygen induced reconstruction. In regime B, the counteracting oxidizing and reducing agents induce dynamic interconversion between simultaneously present oxidized and reduced domains. In regime C, the oxidant dominates and copper is oxidized. The scale bar measures 5 µm. The measurements used to generate the phase diagram are indicated by circles and were performed in the pressure range between 20 and 50 Pa.

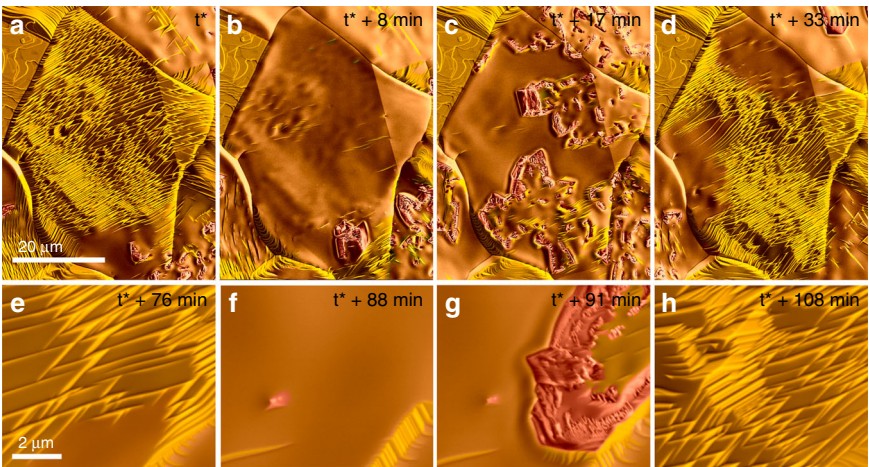

**Fig. 2 Oscillatory redox-dynamics. a–d** Sequence of morphological changes with time as observed at 700 °C in a $H_2/O_2$ atmosphere containing 4% of oxygen. Colours are used to highlight the different stages that are observed during a redox-cycle: **e–h** images recorded from the central grain in **a** at higher magnification under identical experimental conditions during a later redox-cycle.

Fig. 2e–h. They highlight the morphological changes and, together with Fig. 2a–d, the reoccurrence of qualitatively identical features for as long as the gas-phase composition and temperature remain constant. At 700 °C and a total pressure of around 20 Pa, one complete redox-cycle takes about 30 min. The redox kinetics are thus relatively slow. Since the oscillations are not synchronized over the whole surface of the sample under the observed

reaction conditions, and since there are always different surface states co-existing, the locally observed oscillations could not be detected by integral mass-spectroscopy or as temperature oscillations.

As can be seen in Supplementary Movie 4, the transition between different surface states occurs in the form of waves that propagate across individual grains and even across grain

**Fig. 3 Propagating waves. a–c** Wave-like propagation of different surface morphologies. White lines indicate the approximate position of the boundaries between different surface structures. The corresponding movie is available in Supplementary Movie 4.

boundaries. This is shown in Fig. 3 for the transition from the facetted to the flattened surface morphology. At 700 °C in a $H_2$/$O_2$ atmosphere with 4% oxygen partial pressure, these boundaries move at speeds of a few hundred nm/s, depending on grain orientation, direction of propagation and local surface features. Interestingly, oxide growth is exclusively observed on the smooth surface, and the locally observed sequence of (1) surface faceting, (2) surface flattening and (3) oxide growth and (4) oxide reduction is repeated in the same order on all active grains, irrespective of the crystallographic orientation of the surface. Some grains (like the one on the upper left side in Fig. 3) are not showing an oxide growth, although partial changes in surface morphology due to passing waves can be seen. These grains require a higher chemical potential of oxygen for the onset of oxide growth.

The propagation and expansion of oxidation wave-fronts indicates that oxidation is autocatalytic in the sense that oxide growth continues at the growth-front once the potential for oxide growth has built up and oxidation is initiated. The growth speed of the oxide islands is anisotropic and depends on the orientation of the growth-front with respect to the crystallographic orientation of the respective Cu grain (observe the shape evolution of oxide islands in Supplementary Movies 4 and 5). On average, however, the oxidation fronts propagate at speeds similar to the one of the above described moving boundaries between flat and facetted morphologies. As can be seen in Fig. 2g (and also in Fig. 2b, c and Supplementary Movie 5), a bow-wave like structure is running in front of the expanding oxide islands. Due to the action of hydrogen, oxide islands are reduced back to metallic copper. However, reduction only happens after an induction period of, on average, around 2–3 min. Real-time imaging shows that the reduction process is preceded or accompanied by a morphological change of the oxide islands. They develop a well-pronounced facetted, lamellar structure before disappearing (see Fig. 2g, Supplementary Fig. 6, Supplementary Movies 4–6).

In the competition between oxidation and reduction, the size of the formed oxide islands is given by the distance that the oxidation wave-front travels within the induction time for reduction. As a consequence of the propagating oxidation front and the time-delayed reduction, the oxide islands move across the surface in the form of solitary waves. Although oxide growth and reduction are anisotropic, the oxidation and reduction fronts are, on average, moving at similar speeds. This observation is a microscopic expression of the fact that the rates of oxide formation and reduction must be balanced in the bi-stable regime. It is important to note that growing oxide islands do not collide and that oxidation waves are even terminated when approaching each other from different directions. This behaviour is indicative for the existence of an active zone of oxygen capture near the oxidation front[26–28].

Both, an induction period and a transition of the oxide to a reconstructed state have been reported for the case of $Cu_2O$ reduction in $H_2$[29], and similarly, during reduction in CO[30]. Fan

Yang et al.[31] described that CO induced reduction of the $Cu_2O$ surface layer goes through a slow and fast reaction regime. They concluded that chemisorbed oxygen atoms are removed from the $Cu_2O$ surface oxide in the slow regime. As a consequence, the surface reconstructs and an O-deficient $Cu_2O$ phase (oxygen vacancies in a $Cu_2O$ (111)-like layer) forms across the surface, which subsequently leads to a fast step-edge reaction mechanism for reduction. A similar step-edge reduction process was recently observed under reduction in hydrogen[32]. It leads to a retraction motion of atomic steps at the oxide surface and can give rise to pronounced exposure of facets. Such a process could explain the faceting that we see upon reduction, although it occurs here during oscillatory redox transitions, in the presence of hydrogen and oxygen.

**Ex situ structural characterization.** In order to study the redox dynamics in terms of the involved phases and transitions between them in more detail, some in situ SEM experiments were interrupted at points of interest by sudden removal of the reaction gases and rapid cooling. A quenching of the surface dynamics is possible due to the low inertia of the laser heating stage, which allows cooling rates of more than 50 °C/s at high temperatures (above ~500 °C). Cooling is thus very fast compared to the discussed morphological dynamics.

Figure 4a shows an SEM image that was recorded after quenching the redox dynamics on a [110] oriented Cu single crystal (see Supplementary Movie 6) and an image of the cross section through the oxide island that was prepared by focused ion beam (FIB) milling. The darker structures correspond to the oxide islands that are embedded in metallic copper. Figure 4b shows that the Cu underneath the oxide is homogeneous in contrast, indicating the absence of larger voids or buried oxide layers. Some crystallographic orientations as well as the propagation directions of the oxidation and reduction fronts are indicated. One can see that the surface of the oxide reconstructs from a smooth morphology near the growth front to a facetted one with pronounced exposure of $Cu_2O$ 110 facets towards the reduction front (see inset c in Fig. 4b). Furthermore, the formation of the above-mentioned bow-wave at the oxidation front, which can nicely be seen in the Supplementary Movies 4–6, is confirmed.

More details are revealed by analytical TEM investigation of the FIB lamella presented in Fig. 4b. A TEM overview image containing information about crystallographic orientations and locations from where high-resolution images were recorded, is shown in Fig. 5a.

Lattice fringe imaging and electron diffraction show that the structure of the oxide corresponds to $Cu_2O$. The chemical state of copper is further confirmed by the structure of the Cu L-edge in the recorded energy-loss spectrum. It shows the fingerprint of $Cu^{1+}$ in the oxide, and metallic copper in the surrounding (inset of Fig. 5b). From the cross-section of the oxide island, one can see that the oxide layer has expanded into the bulk of the copper foil,

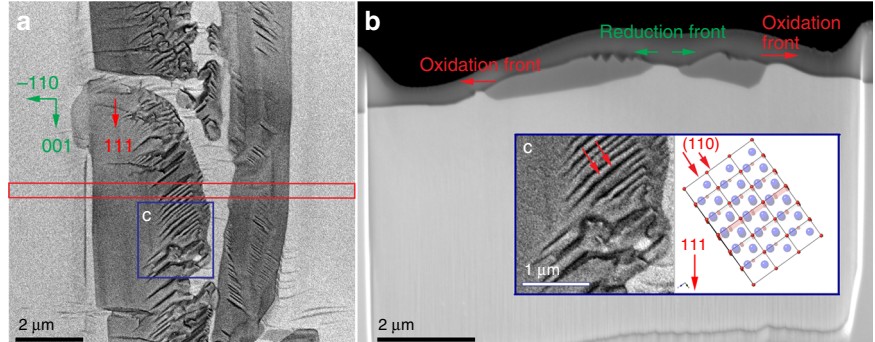

**Fig. 4 Ex situ SEM. a** SEM image recorded after rapid quenching of the redox dynamics. The darker structures correspond to the oxide islands that are embedded in copper metal. A red rectangular indicates the approximate position from where a TEM lamella was abstracted. **b** SEM image of the prepared lamella showing the deposited protective layer for FIB cutting and underneath, the cross-section of oxide islands on the copper. Please note that the oxide island already started to get reduced in the central part at the moment when the in situ SEM experiment was quenched. Red and green arrows indicate the propagation of the oxidation and reduction fronts. c Development of 110 facets is confirmed based on the structural model that is oriented according to the crystallographic orientations abstracted from lattice fringe images of the TEM lamella (see Fig. 5).

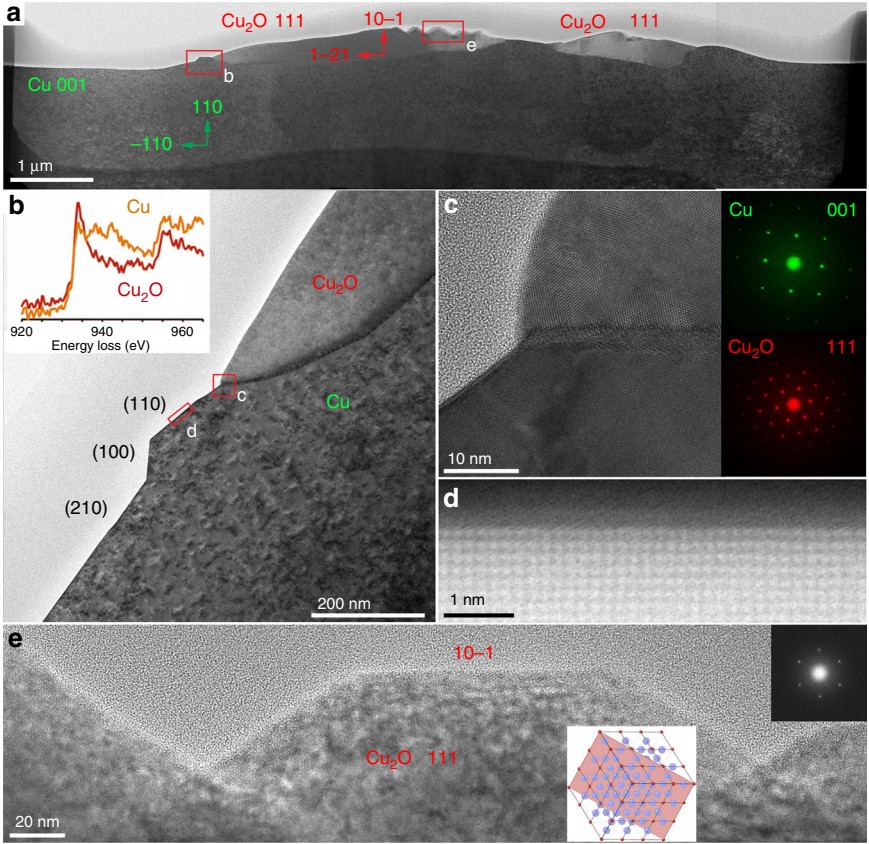

**Fig. 5 Ex situ TEM. a** Overview TEM image of the cross-section that was introduced in Fig. 4. Crystallographic orientations as well as regions from which images in **b**–**e** were recorded are indicated. **b** shows the bow-wave structure at the oxide wave front and EELS spectra of the copper K-edges as inset; **c** HRTEM image of the metal-oxide interface at the oxidation front. Colorized fast Fourier transforms of the lattice fringes with spots due to $Cu_2O$ (red) and Cu (green) are shown as insets. **d** HAADF STEM image indicating that the 110 facet is atomically flat and does not show any obvious signs of reconstruction. **e** Higher magnified image showing the reconstruction of the oxide and pronounced exposure of 110 facets before it is getting reduced. A model of a 111 oriented $Cu_2O$ structure with a coloured 110 plane is shown in the inset.

with the metal-oxide boundary reaching a depth of around 350 nm. Oxide growth in the form of islands instead of a uniform oxide layer as well as the observed logarithmic growth in thickness (see Supplementary Fig. 7) supports earlier reports according to which copper oxidation does not follow the classical Cabrera–Mott model[33,34] under the here considered conditions. Indeed, the latter assumes a uniformly growing film and outward cation diffusion, which would be accompanied by pore-formation in the Cu. Propagation of the metal-oxide interface into the bulk metal, as seen in Fig. 5, suggests oxidation via oxygen interfacial diffusion[27]. Based on the above mentioned observation that growing oxide island do not collide, we further conclude that the kinetics of the oxide formation are dominated by oxygen surface diffusion in the first instance, which is in agreement with earlier

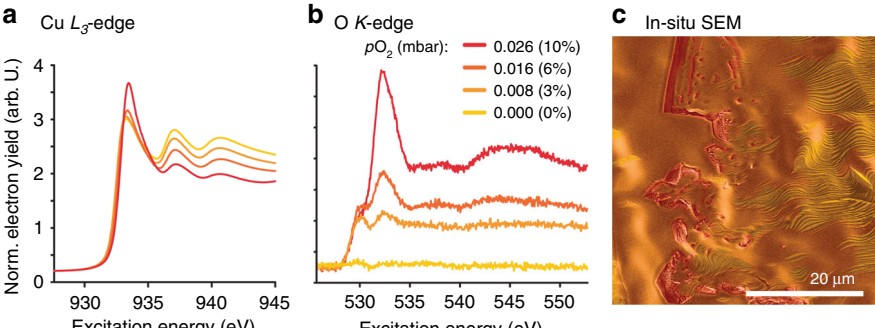

**Fig. 6 In situ NEXAFS.** In situ photoemission measurements of copper foil at 700 °C in a 25 Pa mixture of hydrogen and oxygen, with various oxygen partial pressures ranging from 0 to 10%. **a** Cu L₃-edge NEXAFS spectra and **b** O K-edge NEXAFS spectra. **c** shows an in situ SEM image exhibiting different surface morphologies. The image is coloured to highlight the faceted Cu surface (yellow), smooth Cu surface (orange) and Cu₂O islands (red).

experimental findings by Yang et al.[35] and theoretical descriptions[27].

In Fig. 5b, c, the Cu₂O/Cu interface is inclined with respect to the [001] zone axis of Cu, which is also the viewing direction. Thus, one cannot distinguish a clear atomically sharp interface from the image. However, the existence of sharp boundaries between Cu₂O and Cu with coherent and semi coherent crystalline interfaces has already been reported[36,37] and their formation been observed in situ under low-pressure conditions[38,39].

Figure 5b also reveals some more information about the above-mentioned bow-wave. The convex shape (in all instances, the formation of a hill was observed) and the fact that it is running in front of the oxidation wave indicates that bulging is due to the volume expansion caused by oxidation. The facet of the bow-wave that is facing the oxidation front is the plane on which the oxidation front is propagating. More detailed studies are required to reveal how diffusion processes and oxide growth determine the shape of the bow-wave and how the associated reconstructions are related to a possible precursor phase for oxide growth. So far, we only speculate that bulging in combination with surface diffusion of atoms towards the oxidation front can lead to the formation of a kinetically favoured geometry, which seems to involve a Cu (110) plane[38,39]. The FIB lamella in Fig. 5 was cut perpendicular to both, the (111) plane of Cu₂O and the (001) plane of Cu, respectively. According to the in situ SEM observation, oxide growth is fast along the Cu₂O [111] direction on a Cu (110) surface. As can be seen in Fig. 5a–c, the Cu (110) plane is parallel to the Cu₂O (10−1) plane. We thus see a similar coincidence concerning the (110) planes of Cu and Cu₂O as G. Zhou et al.[40].

One should note that CuO is not observed in the redox regime investigated in this work. The reason is likely that CuO cannot form directly on Cu metal but rather on Cu₂O. Furthermore, at partial pressures of hydrogen considered in this work, conditions are too reducing even for Cu₂O to be stable. Neither are any sub-stoichiometric oxides observed, not in the oxidation, and not in the reduction. Both are direct order–order transitions between Cu and Cu₂O.

Concerning the oxide growth, the crystallographic relations and presence of crystalline interfaces are in agreement with in situ TEM observations by J. Yang et al.[41], LaGrow et al.[39], and with Zhou et al.[38] and we can safely conclude that the crystalline interfaces observed ex situ are not due to rapid cooling in the ESEM and that quenching preserves the state of the interface between metal and oxide. Furthermore, it appears that hydrogen does not have a strong influence on the actual oxide growth (except for changing the kinetics).

In order to confirm this, we have performed oxide growth studies in the absence of hydrogen at reduced oxygen partial pressure. As can be seen in Supplementary Fig. 8, identical features, including the bow-wave, were found. In the absence of H₂, however, the oxidation happens already at much lower pressures (for example, at $8 \times 10^{-3}$ Pa O₂ compared to around 1 Pa O₂ in a 30 Pa H₂/O₂ mixture at 700 °C). We note here that oxide islands do not exclusively nucleate on the flat surface in pure oxygen, but can also nucleate directly on the facetted surface (see Supplementary Fig. 9). Apart from this difference, we find that hydrogen merely shifts the onset of oxidation to higher chemical potentials of oxygen, but does not interfere substantially with the actual oxidation process, once it is initiated. The same is true for the involvement of oxygen in the reduction of the oxide. In both cases, we see direct order–order transitions and qualitative agreement with what is reported in literature for oxidation in pure oxygen[26–28] and reduction in pure hydrogen[29–32], respectively. Information about the interaction between hydrogen and oxygen on copper until the potential for oxidation is reached and vice versa, on the oxide until the potential for reduction is reached, is hidden in the kinetics of the observed reconstructions.

It is not possible to obtain direct information about the adatom–adatom interactions and oxygen chemisorption induced phase transformation from ex situ TEM. That is not only because TEM observation took place in vacuum and at room temperature, but also because the TEM lamella were prepared after quenching the redox dynamics and deposition of a protective surface layer. Concerning the reconstructions, we can thus only conclude that the copper in the facetted and smooth regions is metallic up to the surface. Reliable information about surface coverage and mechanisms that drive the observed wave-like propagation of different reconstructions can only be obtained from in situ methods that are sensitive to the chemical state of the surface under redox conditions.

**In situ NEXAFS characterization.** To investigate the chemical nature of the surface in the redox-state, we turned to in situ near-ambient pressure NEXAFS (near-edge X-ray absorption fine structure). Figure 6a, b show in situ Cu L₃- and O K-edge spectra of a copper foil measured at 700 °C in a 25 Pa atmosphere of H₂ and at various partial pressures of O₂. These are the same conditions used in the ESEM observation. The Cu L₃ spectra show that the surface starts off metallic in pure H₂. When O₂ is added to the feed (pO₂ = 0.8 Pa, corresponding to 3% O₂ in a H₂/O₂ atmosphere) the Cu L-edge indicates metallic Cu, while the O K-edge shows a feature with an edge at 529.5 eV, which is indicative of an O-terminated Cu surface[7].

When the O₂ partial pressure is increased to 16 Pa, signs of Cu₂O begin to appear. Cu₂O is apparent from the Cu L-edge by

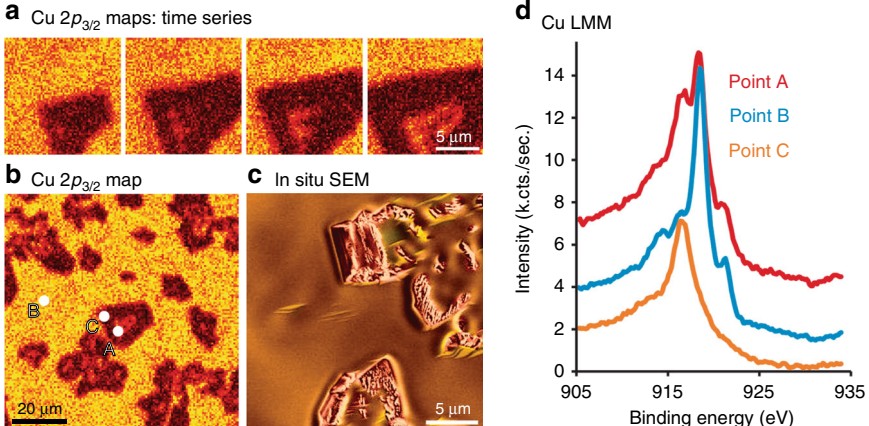

**Fig. 7 In situ SPEM: scanning photoelectron microscopy. a** Maps recorded under 4% $O_2$ and 96% $H_2$ at 650 °C and 20 Pa. The intensity of the photoemission signal within an energy window of 2 eV centred at 932.7 eV was used to generate the maps. **b** Cu surface morphology was frozen by pumping to high vacuum conditions at 650 °C. Three representative points have been chosen to investigate the state of copper: Point A is located at a reduction front, B is located in an area between oxide islands, C on the rim of an oxide island); **c** in situ SEM image for comparison of surface features. **d** Cu LMM spectra corresponding to point A showed a mixture of Cu and $Cu_2O$, point B showed metallic state and point C is $Cu_2O$.

the intensity increase at 932.9 eV (edge position), and in the O K-edge by the formation of a feature at 531.8 eV (edge position)[6,7]. The conditions for these spectra are identical to those in the ESEM measurements for which dynamic redox changes were observed (Fig. 6c). This observation confirms that the morphologies observed in ESEM represent a coexistence of O-terminated Cu and $Cu_2O$. As we have clearly identified the island structures as $Cu_2O$ using TEM, this implies that at least one of the flat or faceted morphologies is an O-terminated Cu surface. Without spatially resolved surface chemical information, one cannot conclusively confirm that different oxygen terminations are the reason for the flat and faceted surface morphologies of copper.

**In situ scanning XPS characterization.** To perform an in situ chemical-state sensitive, microscopic characterization of the material, we utilized micro-focused scanning photoemission microscopy (SPEM)[42,43]. Using this method, we heated a poly-crystalline Cu foil to 650 °C in a mixture of $H_2$ and $O_2$ at 20 Pa. A slightly lower temperature was chosen in order to slow down the kinetics of the process, as SPEM acquisition times are considerably longer than ESEM image acquisition times (ca. 5 min per map for SPEM versus ~0.5 min per image for ESEM). A time series of SPEM maps is shown in Fig. 7a. These maps represent the intensity of the Cu $2p_{3/2}$ photoemission signal within an energy window of 2 eV centred at 932.7 eV. Thus, bright points indicate higher Cu content. Here one can observe the formation of dark islands, which were identified using Cu LMM spectra as $Cu_2O$. The $Cu_2O$ islands grow with time, and eventually, bright intensity forms in the middle of the oxide islands. The bright patches showing up in the middle of the $Cu_2O$ grain in frames 2–4 of Fig. 7a indicates the reduction of $Cu_2O$ to Cu metal. The behaviour of the copper surface in the SPEM set-up nicely reproduces the dynamics that were observed by in situ SEM. This is evident from a comparison of the large-area Cu $2p_{3/2}$ map with an image from the in situ SEM experiments (Fig. 7b, c, respectively). For a better identification of the surface composition, we have measured high-resolution point spectra of the Cu LMM and O 1s regions at locations marked as A, B and C in Fig. 7b. The Cu LMM spectrum at point A and B indicate that Cu in these regions is metallic, while the Cu LMM spectrum taken from point C indicates that it is $Cu_2O$, as expected (see Fig. 7c). Note that spectrum A consists of a mixture of Cu LMM spectra from $Cu_2O$

and Cu because the spatial resolution was not sufficient to isolate a purely metallic region on the oxide grain.

An interesting observation from the in situ SEM investigations was that oxide islands nucleate only on the smooth surface morphology. From the in situ SPEM investigation, we can see that the smooth region surrounding the oxide islands consists of metallic Cu (based on the Cu LMM spectra, Fig. 7d), yet the O 1s spectra of the same regions show the existence of an oxygen species (see Supplementary Fig. 10). Based on the O 1s binding energy and the lack of an oxide signal in the Cu LMM spectra, we conclude that these O-species represent an O-terminated Cu surface. This assignment agrees with the in situ NEXAFS findings. In order to identify differences between facetted and smooth surface, both reoccurring morphologies should be captured on the same grain. So far, we have not been able to attain condition in which both could be sufficiently well distinguished and separated SPEM spectra recorded. However, regions where both smooth and faceted surfaces could be simultaneously observed on two neighbouring grains did not reveal any quantifiable difference in oxygen termination (see Supplementary Fig. 10). This leaves us with the conclusion that indeed, both, flat and facetted morphology, are related to oxygen induced surface reconstructions, without knowing the difference in oxygen coverage.

## Discussion
In the sequence of morphological changes that are observed during one redox-cycle, surface faceting and smoothening always precede oxide formation. In order to understand the reasons for this, we first need to address the difference in surface termination between the facetted and smooth surface. Chemisorbed oxygen is known to induce several surface reconstructions on copper that can give rise to macroscopically observable faceting[44–48]. Experimental and theoretical studies have shown that low-index surfaces pass through different surface reconstructions with increasing oxygen coverage[49–52]. Amongst them are the well-ordered c(2 × 2) phases and the Cu-(2√2 × 2√2) R45°-O missing-row reconstructions on the Cu(100)[52] surface, the p(2 × 1) and c (6 × 2) reconstructions on Cu(110)[49], and the relatively complex '29'- and '44'-structures on Cu(111)[50].

Furthermore, it was shown that $Cu_2O$ nucleation occurs after the surface is covered with an O-termination. For instance, oxidation of Cu(100) proceeds step-wise via the formation of a c(2 × 2) reconstruction, followed by the more O-rich Cu-(2√2 × 2√2)

**Fig. 8 Model.** Illustration of different stages that the surface passes through during a redox cycle at 700 °C at 4% oxygen in a $H_2/O_2$ atmosphere and total pressure of 20 Pa. The time axis is running from the left to the right. $r(O_2)$ and $r(H_2)$ represent the rates of $O_2$ activation and $H_2$ activation, respectively. Arrows indicate kinetic barriers, i.e., transitions that require induction time.

R45°-O reconstruction, and then by the nucleation of $Cu_2O$[34,51,52]. The reconstructions can thus be considered to be the ordered chemisorbed phases preceding bulk oxide growth. It is important to note that some of these reconstructions are relatively stable. In the case of Cu(100), for example, the oxidation from clean Cu to the Cu-$(2\sqrt{2} \times 2\sqrt{2})$ R45°-O reconstruction occurs rapidly, while the further oxidation to $Cu_2O$ is kinetically slower[52].

With respect to the conditions considered in this work, it is furthermore important to point out that no strong pressure dependence of oxidation kinetics was found over a large range of oxygen pressures[34,37,53–55]. For example, Lahtonen et al. showed that an increase of the $O_2$ exposure from $p_{O2} = 8.0 \times 10^{-5}$ to 3.7 Pa only increases the total amount of oxygen on the Cu(100) surface from ~0.56 to ~0.63 monolayers and thus, that oxygen induced reconstructions are quite stable[52,54].

Our reference experiments in pure oxygen were performed in the pressure regime in which the reconstructions are known to be stable (~$10^{-2}$ Pa). As discussed above, we have observed grain orientation dependent faceting as soon as oxygen was introduced into the ESEM chamber (see Supplementary Fig. 2). We have also observed that the reconstructions can change with increasing oxygen partial pressure. The relatively high stability of some reconstructions was reflected by the fact that prolonged annealing in hydrogen at high temperature (up to ~800 °C) was required in order to remove the surface reconstructions (see Supplementary Fig. 2). Based on their dependence on oxygen, their relatively high stability, and on the fact that they are observed before the onset of oxide nucleation, we can safely assign the facetted surface morphologies to macroscopic expressions of oxygen-induced surface reconstructions and associated lowering of surface energy[45,48,56]. They lead, according to our TEM results, to the stabilization of <110>, <100> and <210> planes. In the presence of hydrogen, a higher oxygen partial pressure is required in order to induce such a surface faceting. The characteristics of the reconstructions are, however, similar. Their relatively high stability against further oxidation is reflected by the fact that no direct nucleation of $Cu_2O$ was observed on the facetted surface in the here considered redox regime. Furthermore, the facetted state dominates the redox-cycle and has a longer lifetime compared to the flat state and to the one of the oxide islands[57,58].

Once the kinetical hindrance, which is related to adsobant–adsorbant interaction[54], is overcome, higher oxygen loading induces a transition from the facetted to the flat state. We attribute the flat state to a surface termination with higher O-content for two reasons: Firstly, because crystal shapes tend to be more isotropic at higher oxygen chemical potential due to a larger variety of low-energy reconstructions[55]. Secondly, because the flat state is the one that precedes oxide growth.

Based on the above arguments, we propose a model of the reaction dynamics where the transitions from faceted to smooth to $Cu_2O$ represent oxidation steps and the transitions from $Cu_2O$ through a faceted $Cu_2O$ to a smooth copper and finally, back to faceted copper, represent reduction steps. The involved phases are illustrated in a schematic model in Fig. 8. It is important to note here that similar transitions have been found for the case of

oscillatory CO oxidation by Hendriksen et al., based on operando X-ray diffraction[9] and high-pressure scanning tunnelling microscopy[14].

One can rank the various states described above according to their oxo- and nucleophilicity. The oxophilicity decreases in the order Cu > [O]-Cu > $Cu_2O$. Thus, clean metallic Cu activates $O_2$ the fastest, while $Cu_2O$ activates $O_2$ the slowest. In contrast, the nucleophilicity follows the reverse order, i.e. $Cu_2O$ > [O]-Cu > Cu. Thus, $Cu_2O$ activates $H_2$ the fastest, while Cu activates $H_2$ the slowest. The reason for the complex dynamic and the phase co-existence is that there is no single phase that is good at activating both reactants. Thus, Cu will continue to oxidize until it can activate $H_2$ faster than it can activate $O_2$. This process results in the surface swinging between two extremes, passing through intermediate phases with different oxygen coverage along the way. In fact, the un-terminated, purely metallic state is not explicitly observed here, since the facetted state, induced by sub-monolayer oxygen coverage, is quickly established and more stable. The observed wave-like propagation of different surface states involves surface and bulk diffusion. It is driven by the interplay between autocatalytic processes and kinetic barriers, rather than by variations in the $H_2$ and $O_2$ concentration in the gas-phase above the surface, although the observed propagation of waves across grain boundaries made us first believe in a control by the gas-phase. However, considering the time-scale, it appears that the dynamics are too slow to be related to variations in the gas-phase composition above the surface. Neither do we have indications that they are thermo-kinetic oscillations under the conditions of low conversion studied here. Indeed, the lateral extension of different surface phases with different ongoing processes is in the range of several 10 µm. Copper is a good thermal conductor and temperature gradients would equilibrate fast across these small distances compared with the time scale of the observed oscillations.

Finally, we return to the question regarding the dominant mechanism of $H_2O$ formation. In order to do so, we compare the morphological evolution of the surface during a redox cycle with what is observed during slow oxidation in pure oxygen (at ~$10^{-2}$ Pa and 700 °C). The first and most obvious difference is that oxide islands get reduced in the presence of hydrogen, while they keep growing until the complete surface is oxidized when exposed to pure oxygen. The second difference lies in the sequence of surface reconstructions. While oxide islands can directly nucleate on the facetted surface in pure $O_2$, the co-exposure of $H_2$ and $O_2$ leads to redox cycles. In the regime discussed here, these redox cycles involve a pronounced phase during which the surface remains in the flat state. It is oxygen terminated according to the SPEM results and emerges after the quite rigid and stable facetted surface breaks up. The persistence of the flat morphology in the presence of hydrogen is a consequence of a delayed oxide nucleation. The state of the flat surface can therefore be associated to a frustrated phase transition[59]. Its feature-less morphology is indicative for a high mobility of surface species. Indeed, the above-mentioned zone of oxygen capture around each oxide island, as well as the propagation of a bow-wave in front of the oxide islands, indicate a high mobility of oxygen and copper

species. In this state, the capturing of oxygen from the surface by hydrogen runs in competition with oxide nucleation. It is responsible for a delayed or, at low oxygen partial pressure (Regime A), even suppressed oxide nucleation. We thus speculate that it is this highly dynamic flat state on which a Langmuir–Hinshelwood- or Eley–Rideal-like process leads to catalytic water formation.

Due to its high oxygen affinity, metallic copper can only be maintained at very low oxygen partial pressure. Water formation on metallic copper is thus always oxygen limited (regime A in Fig. 1). In the intermediate regime, where metallic and oxidized domains are simultaneously present, water formation proceeds through both, the above-mentioned competing reaction to oxide formation, and the reduction of continuously forming oxide islands. At higher oxygen concentrations (above 25–35% in the here considered temperature and pressure range), the whole copper surface will be oxidized and basically inactive (regime C in Fig. 1). Copper is thus too oxyphilic for being a good catalyst for hydrogen oxidation. Since metallic copper gets increasingly stable at higher temperature, it is possible to shift the boundary between regime B and C to higher oxygen concentrations, and thus, to obtain higher hydrogen conversion.

Overall, the observed behaviour of copper under hydrogen oxidation reflects the response of a single phase to the presence of two complementary components in a gas-phase. The preferential reactivity towards one of the reactants induces a response of the catalyst that in effect switches the reactivity towards the other reactant. The gas-phase induced phase change acts as delayed feedback mechanism and drives the system into a dynamic equilibrium that is defined by oscillatory behaviour and emergence of complex spatio-temporal dynamics with alternating phases and surface terminations at any point in time. Since the oscillatory behaviour implies that the surface repeatedly and sequentially passes through different phases, this provides a mechanism in which transient active sites can be re-formed in each cycle. If the system can be driven in a bi-stable state, activity arises at the phase boundary and in the transition between the co-existing phases[59]. This interpretation is in line with observations made in electrocatalysis, where similar evolution of the structure and phase composition as a function of the applied potential and composition of the electrolyte has been reported for Cu catalysed $CO_2$ electroreduction[60,61]. The presence of $Cu^+$ species on the surface during the reaction was found to be key for the catalytic performance also in methanol oxidation[6], ethylene epoxidation[7] and CO oxidation[12]. Since phase co-existence and -cooperation has been experimentally observed[62], and predicted by theory for several other systems[63–65], the emergence of activity near a phase-boundary and the involvement of frustrated phase transitions might be a general characteristic for redox-active catalysts. Static pictures are then insufficient to describe catalytic function.

In conclusion, we have presented an in situ study on the dynamic behaviour of metal catalysts in redox reactions using hydrogen oxidation on copper as model system. The combination of in situ imaging and spectroscopic tools provides new and direct insights about the dynamic state of an active catalyst. It was shown that the counteracting action of the oxidant and reductant with respect to the phase stability of copper drives the system into a dynamic equilibrium that is characterized by oscillatory redox phase transitions and a constant interconversion between coexisting metal and oxide islands. We have identified the involved phases as $Cu_2O$ and two surface terminations of metallic copper, which all differ in their oxygen chemical potentials. The oscillatory dynamics are a consequence of the fact that a single-phase catalyst cannot be equally reactive towards two complementary components in the gas-phase. In contrast to the conventional wisdom of static active sites being produced during catalyst synthesis, the process of dynamical restructuring near a phase boundary provides a convenient mechanism for a continuous generation of short-lived high-energy sites for catalytic action. The visual information provided here is generally missed by laterally averaging in situ spectroscopy methods and can now be used to refine or confirm earlier assumptions about the state and composition of the active surface. The insights presented in this work are of importance for theoretic modelling. They highlight the fact that the reaction conditions not only activate the reactants, but also the catalyst. Static pictures are thus insufficient to describe catalytic function and the art of catalyst design and operation is to stabilize the relevant dynamic state. By providing information about the relevant kinetic barriers and mechanisms that are involved in the emergence of oscillatory dynamics, this work provides insight for modelling based on microscopic understanding and kinetic Monte Carlo simulations. Finally, the work highlights the importance of combing complementary laterally resolving in situ methods that are executed under the same reaction conditions.

## Methods

**In situ SEM**. In situ scanning electron microscopy measurements were performed using a commercial ESEM (FEI Quanta 200). The base-pressure of the instrument is $2 \times 10^{-5}$ Pa, with residual gas composed mainly of $H_2O$, $N_2$ and $O_2$. The instrument is equipped with a home-built infrared laser heating stage, oil-free pre-vacuum pumps and a gas supply unit with mass flow controllers from Bronkhorst. Polycrystalline copper foils of 0.1 mm thickness and 99.998% purity were purchased from Advent Research Materials Ltd. Prior to all experiments, copper samples were cleaned using ion polishing (Gatan Model 691, 1 h @ 5 kV $Ar^+$), followed by annealing in 20 Pa high purity (99.999%) $H_2$ at 700–800 °C inside the differentially-pumped chamber of an environmental scanning electron microscope (ESEM) for 2 h (see Supplementary Fig. 1 and Supplementary Movie 1). EDX elemental mapping was directly performed in the ESEM after cooling to room-temperature using a Si (Li) detector from Bruker.

During in situ measurements, reaction gases were directly fed into the chamber of the microscope. At a hydrogen flow of 10 sccm and oxygen flows between 0 and 5 sccm, the pressure in the chamber equilibrated in the range between 20 and 50 Pa. The temperature was measured via type K thermocouples that were directly spot-welded onto each sample. Images were recorded using the gaseous secondary electron detector[66]. At each set temperature, the temporal evolution of the surface morphology was monitored for at least 2 h in order to make sure that the system was given enough time to adapt to the changed conditions and reach a dynamic equilibrium. Evaluation of the effect of the electron beam was performed by comparing the structural dynamics recorded at different magnification, different acceleration voltage and dose rate. Constant image recording and beam on/off experiments, as well as observations at different magnification, beam currents and pixel dwell-time, showed the same processes and morphological changes on the surface and no influence of the electron beam.

**In situ NEXAFS**. Near-ambient X-ray absorption fine structure (NEXAFS) measurements were carried out at the Innovative Station for In Situ Spectroscopy (ISISS) beamline at the Helmholz-Zentrum Berlin (HZB) synchrotron light source (BESSY II). The NAP-XPS set-up is equipped with a Specs GmbH Phoibos 150 differentially-pumped electrostatic lens and analyser system. It contains a reaction cell in which the sample is mounted and enables measurements at pressures of up to 100 Pa. Details of the equipment are available elsewhere[8]. During in situ experiments, reaction gases (from Westfalen AG, purities: hydrogen and oxygen 6.0 N) were continuously fed into the reaction cell via mass-flow controllers. The pressure in the chamber was maintained at 20 Pa during the measurements. Samples were heated from the back side using an infrared laser, and the temperature was measured by type K thermocouples that were mechanically clamped onto the surface of the sample.

**In situ SPEM**. Environmental scanning photoelectron microscopy (ESPEM) measurements were performed at the ESCA microscopy beamline at the Elettra synchrotron facility in Trieste, Italy. The setup consists of a hemispherical energy analyser, attached to a chamber that contains a specially designed cell that can be back-filled with a reaction gas mixture to 20 Pa. The reaction cell is separated from the high vacuum region via a 300 μm diameter aperture. The X-ray radiation used as the excitation source was generated via an undulator and focused to a 190 nm-diameter spot on the surface of the sample using Fresnel optics. The photon energy used for the measurements was 1071 eV. The sample was heated during the measurements using a boron-nitride-coated resistive heater. Further details of the experimental set-up can be found elsewhere[43].

**FIB milling and TEM analysis**. Samples were transferred to a Heilos G3 focused ion beam (FIB) SEM for target preparation of TEM lamella. High-resolution transmission electron microscopy (HRTEM), high angle annular dark-field (HAADF) scanning TEM (STEM), electron energy-loss spectroscopy (EELS) and EDX mapping on TEM lamella was performed using a double corrected JEOL ARM 200F instrument that is equipped with a Gatan Quantum ER imaging filter and a JEOL silicon drift EDX detector.

## Data availability
All recorded images and analysed datasets for this work are available from the corresponding author on reasonable request.

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

## Acknowledgements

J.C. is grateful to the CSC (Chinese Scholarship Council) for financing her PhD at the FHI. J.C. and M.W. want to acknowledge Gisela Weinberg for her assistance in ESEM training and Danail Ivanov for sample preparations. C.C. acknowledges funding through the ETH Research Grant ETH-36 18-2.

## Author contributions

M.W. conceived and planned the research. He supervised J.C. during her PhD thesis and analysed the T.E.M. data and most of the in situ SEM data. J.C., M.G. and M.W. wrote the first version of the manuscript. M.W. rewrote the manuscript to its final version. J.C. conducted the in situ S.E.M. experiments. A.R. and C.C. conducted supporting in situ S.E.M. experiments. M.P., X.H. and E.W. conducted TEM experiments for this work. A.H. prepared the FIB lamella. S.H. measured EBSD grain orientation maps that supported this work. S.B., L.G., J.C. and M.G. performed the in situ NEXAFS and in situ SPEM experiments and analysed the data. R.S. and M.A. participated in the discussion of results and enabled this work. M.G. co-supervised J.C. during her work and wrote the in situ NEXAFS and in situ SPEM part of the manuscript.

## Competing interests

The authors declare no competing interests.
