## [Peer Review File · Nature Communications]

Editorial Note: This manuscript has been previously reviewed at another journal that is not operating a transparent peer review scheme. This document only contains reviewer comments and rebuttal letters for versions considered at *Nature Communications* .

REVIEWER COMMENTS

Reviewer #1 (Remarks to the Author):

The manuscript by Cao et al. reports the real-time monitoring of dynamic evolution of Cu surfaces in response to the imposed environmental stimuli. It is demonstrated that the competing action of simultaneously present O₂ and H₂ gases results in oscillatory redox phase transitions. While it is not surprising that both oxidation and reduction occur simultaneously for the H₂+O mixture, the wealth of the surface dynamics observed from the in-situ imaging under the reaction conditions is interesting. The revealed mechanism for maintaining the dynamic equilibrium via the oscillatory redox reactions is particularly appealing. These results provide new insight into understanding the dynamic nature of surfaces under complex reaction conditions and are of broad relevance to various catalytic reactions.

The authors have addressed my concerns raised from the previous review round. I would suggest the authors to add more information regarding "constant re-generation of transient active sites". The surface species involved in the reaction include metallic Cu (Cu⁰), Cu⁺, chemisorbed O, and lattice oxygen (O₂⁻). It lacks some clarity about which surface species are the catalytically active phases, how these active species (and sites) are constantly regenerated from the oscillatory dynamics, and how the non-active species can be minimized by utilizing the oscillatory kinetics in order to promote the reactivity and selectivity.

Reviewer #2 (Remarks to the Author):

In the revised submission the authors replied appropriately to my previous comments but did not make any changes in the main text. I understand because they can not provide experimental evidence. Although the data are beautiful and the proposed model is qualitatively self-consistent, I require that the authors must add a paragraph in the main text to discuss the factors raised by the referees that they can not exclude so that the readers will acquire comprehensive information, instead of (likely) misleading information. After that, I can recommend the publication of the submission in Nature Communication.

Respected Reviewers,

Your reviewing during the first round was critical, but also motivating and constructive. We felt that you find interest in our work and are willing to help us to improve it, and through that, to finally make our results visible and accessible to a large community. You have demonstrated your positive attitude once more by accepting a second round of reviewing. We sincerely thank you for your time and efforts.

In the following, we try to respond to all of your remaining comments:

Reviewer #1:

We are happy to read that: *“The authors have addressed my concerns raised from the previous review round.”*

You state: *“I would suggest the authors to add more information regarding “constant re-generation of transient active sites”.*

Our response:

Following this suggestion, we have added the following statement regarding the constant re-generation of transient active sites to the discussion on page 21: *“Since the oscillatory behavior implies that the surface repeatedly and sequentially passes through different phases, this provides a mechanism in which transient active sites can be re-formed in each cycle.”*

Reviewer #1:

“The surface species involved in the reaction include metallic Cu (Cu⁰), Cu⁺, chemisorbed O, and lattice oxygen (O₂⁻). It lacks some clarity about which surface species are the catalytically active phases, how these active species (and sites) are constantly regenerated from the oscillatory dynamics, and how the non-active species can be minimized by utilizing the oscillatory kinetics in order to promote the reactivity and selectivity.”

Our response:

The clarity about which surface species are the catalytically active species cannot explicitly be delivered. It is, first of all, a concerted process in which not just one species can be considered as sole responsible active entity. In the manuscript, we describe the specific role of the “flat” surface state that precedes oxide formation and, according to the in-situ XPS, contains chemisorbed oxygen. It is in this state, where the presence of hydrogen delays oxide formation by harvesting some of the chemisorbed oxygen from the surface. We feel that we should not make any further, more explicit statements based on the data we have. We are looking forward to future work that can, hopefully stimulated by this work, shine more light on detailed atomistic dynamics in the seemingly flat surface state: how the oxygen concentration builds up near the oxide growth front, and how hydrogen influences the attempt rates at the growth front of the oxide. We also do not want to speculate about any means by which the non-active species can be minimized by utilizing the oscillatory kinetics to promote the reactivity and selectivity. We hope that reviewer #1 meant this statement rather as

a comment. However, motivated by the reviewer's comment, we have moved an important sentence from the discussion to the conclusion on page 22: "...the art of catalyst design and operation lies in stabilizing the relevant dynamics." We think that this is an important conclusion, especially for those in the field that believe in the picture of a static catalyst.

Reviewer #2:

"In the revised submission the authors replied appropriately to my previous comments but did not make any changes in the main text. I understand because they cannot provide experimental evidence. Although the data are beautiful and the proposed model is qualitatively self-consistent, I require that the authors must add a paragraph in the main text to discuss the factors raised by the referees that they cannot exclude so that the readers will acquire comprehensive information, instead of (likely) misleading information. After that, I can recommend the publication of the submission in Nature Communication."

Our response:

For us it was first not clear, which points Reviewer #2 wants us to explicitly add in the form of a paragraph into the manuscript. So, we went back to the points addressed in the previous review round:

The first point was: *"The model reaction, H₂ oxidation at 700 degree C catalyzed by Cu, is really not typical for heterogeneous catalytic reactions. I assume that the choose of such a model reaction is to satisfy the resolutions of ESEM. Thus, how common the observed complex spatio-temporal dynamics in the catalyst composition is in catalytic reactions is doubtful."*

We think that we have answered the first part of this question with our response in the last reviewing round. In the manuscript, we show that the redox-dynamics are observed across a large temperature regime and we state that extended lateral synchronization of the complex dynamics was observed in a certain regime of H₂/O₂ ratio and temperature. We have studied this regime, because it is more understandable and less chaotic than the other regimes, as explained in the manuscript. In the text, there is already a statement that one of the reasons for the higher temperatures that we also cover is related to the lower pressure at which our in-situ SEM works compared to industrial conditions.

Regarding the question about how common the observed complex spatio-temporal dynamics are, we can refer to many published papers on oscillatory dynamic by Ertl and co-workers and review articles, for example by F. Schüth ("Oscillatory Reactions in Heterogeneous Catalysis", *Advances in Catalysis*", Volume 39, 1993, Pages 51-127) or R. Imbihl and G. Ertl ("Oscillatory Kinetics in Heterogeneous Catalysis", *Chem. Rev.* 1995, 95, 697-733). Regretfully, we realized that we forgot to cite two papers that we consulted during the writing of the manuscript: "Reaction pathways in methanol oxidation: kinetic oscillations in the copper/oxygen system" by H. Werner, D. Herein, G. Schulz, U. Wild & R. Schlögl, published in *Catalysis Letters* volume 49, pages 109–119(1997) and „Oscillating Oxidation of Propene on Copper Oxides" by A. Amariglio,

O. Benali and H. Amariglio, Journal of Catalysis. We have added a reference to these papers at the place where we first describe the appearance of oscillatory behavior in our experiments on page 6: “This is interesting, since oscillatory behavior has also been observed for copper in methanol and propane oxidation.”^{33, 34}”

Another point raised by Reviewer #2 in the first round of reviewing:
“No reaction data is provided, thus it is unknown that the observed complex spatiotemporal dynamics in the catalyst composition results in a steady-state catalytic reaction or an oscillating catalytic reaction.”

We have now added the following statement in page 8 of the manuscript:
“Since the oscillations are not synchronized over the whole surface of the sample under the observed reaction conditions, and since there are always different surface states co-existing, the locally observed oscillations could not be detected by integral mass-spectroscopy or as temperature oscillations.”

3) Did the authors consider the likely effect of local temperatures fluctuations caused by the reaction heats on their experimental observations? The reactions proceed at near ambient pressure and the reaction heats should be large enough to affect the local temperatures of different parts of the copper catalyst. For example, a Cu region undergoing oxidation reaction is likely with a different temperature from a Cu₂O region undergoing reduction reaction. The temperature strongly affects the reaction kinetics, thus the local temperature fluctuations accompanying the catalytic reactions can result in an oscillating change in the local catalyst areas. Actually, this is quite known that low-pressure reaction conditions and thin metal single crystals were used to study the oscillatory catalytic reactions in order to avoid the temperature effect.

Thank you for this comment. We thought about effects of temperature, but concluded that the dynamics are far slower than the time-scale required for equilibration of temperature fluctuations: One complete oscillation (stepped oxygen terminated metal → flat metallic state with surface oxygen → oxide growth → oxide reduction) takes around 20-30 minutes. The lateral extension of different phases is in the range of several 10 μm. Copper is a good heat conductor; temperature differences would equilibrate faster than the dynamics. Further studies are needed to definitely clarify what defines the speed of the different transitions and associated barriers, but it is most likely related to diffusion of vacancies and stability of surface configurations. In order to take this point into account, we have added the following sentences into the discussion on page 20:

“Neither do we have indications that they are thermo-kinetic oscillations under the conditions of low conversion studied here. Indeed, the lateral extension of different surface phases with different ongoing processes is in the range of several 10 μm. Copper is a good thermal conductor and temperature gradients would equilibrate fast across these small distances compared with the time scale of the observed oscillations.”

We think that we have answered the question 4 of Reviewer #2 already in the first round of reviewing and think that no addition or changes to the manuscript is required for this point.

Finally, with the above indicated modifications and answers to the Reviewers, we hope that all can agree now that this work is suitable for publication in Nature Communications.